# A Qualitative Study on Frontline Nurses’ Experiences and Challenges in Providing Care for COVID-19 Patients in the Volta Region of Ghana: Implications for Nursing Management and Nursing Workforce Retention

**DOI:** 10.3390/healthcare11071028

**Published:** 2023-04-04

**Authors:** Peter Adatara, Anthony Kolsabilik Kuug, Felix Kwasi Nyande, Ellen Eyi Klutsey, Beatrice Bella Johnson, Martin Kwabena Nyefene, Philemon Adoliwine Amooba, Francisca Achaliwie, Raymond Saa Eru Maalman, George Boni Sedinam, Emmanuel Barima Agyemang Prempeh, Matilda Mawusi Kodjo

**Affiliations:** 1Department of Nursing, School of Nursing and Midwifery, University of Health and Allied Sciences, Ho PMB 31, Ghana; 2Express Medical, 2180 Antwerp, Belgium; 3Department of Nursing, Faculty of Allied Health Sciences, College of Health Sciences, Kwame Nkrumah University of Science and Technology, Private Mail Bag, University Post Office, Kumasi, Ghana; 4Department of Maternal and Child Health, School of Nursing and Midwifery, CK Tedam University of Technology and Applied Sciences, Navrongo P.O. Box 24, Ghana; 5Department of Basic Medical Sciences, School of Medicine, University of Health and Allied Sciences, Ho PMB 31, Ghana; 6Department of Accident and Emergency, Ho Teaching Hospital, Ho P.O. Box MA 374, Ghana; 7School of Allied Health Sciences, University of Health and Allied Sciences, Ho PMB 31, Ghana; 8Department of Nursing, Ho Polyclinic, Ho P.O. Box HP 68, Ghana

**Keywords:** frontline nurses, experiences, challenges, providing care, COVID-19

## Abstract

Since the outbreak of COVID-19, studies related to the COVID-19 pandemic have been published widely. However, there are limited qualitative studies in Ghana that explored and shed light on frontline nurses’ experiences and challenges in caring for patients diagnosed with COVID-19. This study aimed to explore frontline nurses’ experiences and challenges of providing care for COVID-19 patients in the Volta Region of Ghana. This study adopted a descriptive qualitative research design to collect data. We conducted the study among frontline nurses who provided nursing care for COVID-19 patients in the Treatment Centre for COVID-19 cases in the Volta Region of Ghana. A purposive sampling method was used to select fifteen (15) participants for this study. We collected data through individual in-depth interviews facilitated by a semi-structured interview guide. The content analysis approach was used to analyse the data. The results showed: frontline nurses received inadequate information and training during the initial stages of the pandemic; stress and burnout because of inadequate staffing; logistical challenges; stigmatisation by family members and friends; frontline nurses displeased with the decision to exclude other nurses as frontline workers; participants made some recommendations towards supporting frontline nurses for effective management of patients during pandemics. This study revealed an in-depth understanding of the experiences of frontline nurses who provided nursing care to COVID-19 patients during the pandemic. Our study concludes that the frontline nurses experienced both physical and psychological problems while caring for COVID-19 patients at the treatment centre. Some challenges frontline nurses encountered were inadequate information on COVID-19 prevention and management in the early stages of the pandemic, logistical inadequacies, and stigmatisation in providing care for COVID-19 patients during the pandemic, all of which affected the quality of nursing care, work productivity, and efficiency. Therefore, nurse managers need to provide support to frontline nurses providing care for patients with COVID-19.

## 1. Background

Since the outbreak of novel coronavirus 2019 (COVID-19) in Wuhan, China, towards the end of December 2019, as in many pandemics, nurses have been at the forefront in the fight against COVID-19 [1,2]. During the COVID-19 pandemic, frontline nurses who were closely involved in providing care for patients experienced many physical and psychosocial difficulties [3,4]. The World Health Organization (WHO) reported that the global death toll resulting from the COVID-19 pandemic was nearly five million and that the number of cases was over 240 million as of October 18, 2021 [5]. The WHO estimates that between 80,000 and 180,000 healthcare workers, including nurses and midwives, could have died from COVID-19 between January 2020 and May 2021 [5]. The International Council of Nurses (ICN) reported that over 1500 nurses have died in 44 countries since the outbreak of the COVID-19 pandemic [6,7].

In Ghana, according to the Ghana Health Service (GHS), over 2000 health care workers have tested positive for COVID-19, with over 840 of them being nurses and midwives. Out of the total number of nurses and midwives who tested positive for COVID-19, five nurses and midwives had died from complications related to COVID-19 as of the time of this study [8]. The Ghana Health Service also reported that the increasing numbers of health care workers being infected with COVID-19 could be attributed to the inadequate and erratic supply of personal protective equipment (PPE) to health workers, both in quantity and quality, as well as a general non-adherence to infection prevention and control protocols in most facilities [8]. Reports showed that delays in the receipt of COVID-19 test results for patients being managed in wards increased the exposure of health personnel to the virus [9].

Health and care workers, including nurses, are the foundation of health systems and the driving force in combating the COVID-19 pandemic [1]. Nurses play significant roles at the frontline in controlling the pandemic and providing patient care during the pandemic. As frontline workers during the pandemic, as well as nurses and midwives who come in contact with COVID-19 patients for a long period, predisposing them to contracting the virus and infecting their family members [9]. Study results from other countries show nurses experienced psychological problems such as depression and restlessness due to long hours of providing nursing care to COVID-19 patients [10]. Research studies reported that many nurses contracted COVID-19 and infected their families [11,12]. This ultimately affects the quality of life of nurses negatively, which is a concern for governments and other stakeholders in healthcare systems. Protecting frontline nurses and midwives from infection and mortality must be a core element of any pandemic response because of the critical role they play during pandemic responses.

Although frontline nurses played a significant role in providing nursing care during the COVID-19 pandemic in Ghana and some lost their lives to the disease, there are still limited scientific studies within the Ghanaian context to understand the experiences of nurses at the forefront of providing care during the COVID-19 pandemic. The few studies that explored the experiences of frontline healthcare workers who provided care during the COVID-19 pandemic [13,14,15,16,17] were conducted outside the Ghanaian context [17,18,19,20,21,22,23,24]. Therefore, there is a need to conduct a study on the experiences and challenges of frontline nurses in Ghana who are providing care for COVID-19 patients. This study will provide a platform for nurses to share their experiences, challenges, and insights on providing care for COVID-19 patients. It will also allow the identification of the specific challenges that nurses face and how they can be addressed to improve the quality of care provided for COVID-19 patients in Ghana.

## 2. Methods

### 2.1. Research Design

This study adopted a case study research design to collect data. Case study design is a research method that involves the detailed examination of a particular case, such as an individual, group, or organization, in order to gain a deep understanding of a phenomenon [25]. This design enabled the researchers to gain an in-depth understanding of the experiences of frontline nurses who provided care to COVID-19 patients during the pandemic. The design allowed the frontline nurses to fully describe their experiences managing COVID-19 in a developing country such as Ghana. In conducting this study, we adhered to the criteria for reporting qualitative research (COREQ) and acceptable practices in fieldwork, analysis, and interpretation.

### 2.2. Participants and Setting

Participants for this study were chosen using the purposive sampling technique. The purposive approach was used to choose participants with similar experiences in accordance with the broad criteria established by the researchers. This made it possible for the researchers to interview as many people as possible who had first-hand knowledge of caring for COVID-19 patients throughout the pandemic. The sample size was determined by the saturation of the data, a point in the research process when no new information was being discovered [25]. In this study, we reached the saturation point with 15 nurses. The inclusion criteria were:Nurses who personally provided nursing care to COVID-19 patients at the treatment facility during the pandemic;Nurses who agreed to participate in the research.


The criteria for exclusion were:
Healthcare professionals who did not hold valid nursing and midwifery licenses;During the pandemic, nurses who did not directly offer nursing care to COVID-19 patients at the treatment centre;Nurses who declined to participate in the research.

### 2.3. Data Collection

We collected the data through individual, semi-structured interviews facilitated by a semi-structured interview guide from October 2020 to January 2021. The semi-structured interview guide was prepared based on previous studies. We held interviews over the phone because of the COVID-19 pandemic. The major reason for this mode of data collection was to avoid exposing research participants and researchers to COVID-19. The interviews were all conducted in English by the first author (P.A.), who has a PhD in nursing and has also published some qualitative research articles. We used the Director of Nursing as the gatekeeper because she was the one responsible for selecting and assigning nurses to work as frontline nurses at the COVID-19 treatment centre.

Prior to the interviews, the participants were provided with relevant information about the study. This included information about the purpose of the study, the voluntary nature of the interview, participants’ rights, the interview process, and the audio recording of the interview. We mutually agreed on the date and time of the interview between the researcher and participants. After obtaining the sociodemographic information, the interview guide was used to facilitate discussion and ensure that we pursued the same basic lines of inquiry. We asked every participant an opening question: “What are your experiences of providing care for COVID-19 patients during the pandemic?”. Further questions allowed the participants to explain their experiences and encouraged them. We encouraged participants to dialogue about their experiences providing care for COVID-19 patients. The duration of the interview varied between 30 and 60 min, depending on the participant. After the interview, we performed a summary of the interaction, highlighting key points to validate the data through member checking [26]. We shared daily interviews with other authors to review and provide feedback on the process. This iterative approach strengthened the data elicitation process. Interviews continued until we achieved data saturation. Each day, we wrote comprehensive field notes. We incorporated these field notes into the data set. We interviewed 15 nurses.

### 2.4. Data Analysis

Data analysis was performed using content analysis. The interviews were transcribed verbatim and validated by the participants. The researchers adopted an open-coding approach, where they read and examined the data to identify concepts and themes. We read through the transcripts several times to familiarize ourselves with the data.

We analysed concurrent data without pre-empting themes to guide this study. Three authors, who have PhDs in nursing and experience in qualitative research methods, manually conducted the data analysis. They did this to ensure the trustworthiness of the analysis, as using multiple coders ensures intercoder reliability. The researchers first read and reread the transcripts several times to construct meaning out of frontline nurses’ experiences of providing care for COVID-19 patients during the pandemic. We coded the transcripts after carefully reading the sentences word by word to identify words or phrases that spelled out the meaning of the sentences. Before and during the coding process, the three authors coded the same interviews to identify and discuss differences and check for inter-coder reliability. We categorised similar words and phrases to form themes and ensure that they represent the nurses’ views. The researchers discussed and agreed on the themes. Direct verbatim quotes from the nurses were used to support the findings.

### 2.5. Trustworthiness of the Study

The trustworthiness of this study was ensured by using the same interview guide to interview all participants in the study. The lead author, who has a PhD and many years of experience with qualitative research studies, conducted all of the interviews. Some of the participants were called back later, when the data were being transcribed and analysed, to ask questions for clarification as a form of member checking to ensure the trustworthiness of the results. An audit trail was used to see if the conclusions, interpretations, and recommendations could be linked back to the source of the information or data. Field notes that were taken during data collection were used as backups during transcription to verify participants’ responses. To add to the reliability of the results, the researcher recorded and transcribed all of the interviews word for word. Concurrent data collection and transcription ensured that emerging themes and sub-themes were probed in subsequent interviews.

### 2.6. Ethical Consideration

The research had ethical approval from the Ghana Health Service Ethics Review Committee (GHS-ERC031/011/20). We sought written permission from the Ho Teaching Hospital. We obtained informed consent from the research participants after explaining the content of the information sheet to each of them, including the purpose and aim of the research study. The participants digitally signed the consent forms.

## 3. Results

### Characteristics of Participants

The results in Table 1 showed that the majority, 10 (67%), of the participants were female nurses, while 5 (33%) were male nurses. The participants’ ages ranged from 26 to 50. All participants had a bachelor’s degree as their highest qualification. The majority (11) of the participants were married, while only 4 (27%) had never been married. The highest rank among the participants was that of a Principal Nursing Officer (PNO), while the lowest rank was that of a Nursing Officer (NO) as shown in Table 1.

We identified the following major themes from the data analysis:I.Inadequate information and training about COVID-19 and its management.II.Stress and burnout among frontline nurses because of workload.III.Stigmatisation by family members and friends.IV.Logistical challenges.V.Frontline nurses were not happy about the exclusion of some nurses from being frontline workers.VI.Recommendations towards the prevention and management of COVID-19.


**Theme 1: Inadequate information and training on COVID-19 management.**


Training on COVID-19 prevention and management is extremely vital for frontline healthcare workers, particularly nurses who provide direct care to COVID-19 patients in healthcare facilities. Participants in this study expressed the importance of training in the safe use of personal protective equipment (PPE) as well as more general training about COVID-19 in the effective management of the COVID-19 pandemic. However, some participants reported they did not have adequate information and training on COVID-19 at the initial stages of the pandemic.


*“When we first started caring for COVID-19 patients as nurses in the early stages of the pandemic, we didn’t know much about the safety rules and how to use PPE safely. They did not train us when we started working at the COVID-19 treatment center. I was very unhappy with the conditions under which we were supposed to work.”*
 (PA 02).

Some of the people in the study talked about how they did not receive enough training and were not ready at the start of the pandemic to be frontline nurses, but they improved as the work went on.


*“I did not get adequate training and was not well prepared at the beginning of the pandemic to provide care as a frontline nurse.” “But I picked up as work progressed, although that was not without making mistakes along the way.”*
 (PA 03).


*“Very unprepared. It had not taken me through any formal training at the beginning of the pandemic; I was just verbally told what to do and how to wear the PPEs available. However, recently there have been courses online on the COVID-19 virus.”*
 (PA 04).

The majority of the participants attributed the inadequate information and training at the beginning of the pandemic to the fact that the health facilities were not adequately prepared to handle COVID-19 outbreaks.


*“I think we were not given adequate information and training on COVID-19 because the health facilities were not adequately prepared to fight COVID-19, and as a result, frontline workers were recruited to work before they were given training and information on how to prevent or manage it.”*
 (PA 07).


*“I must say that my health facility was not prepared. They did not know what to do in the early stages of COVID-19. Therefore, we started working as frontline workers before information concerning how to prevent and manage it started coming to us.”*
 (PA 09).

Participants from these health facilities reported a lack of adequate information and training on COVID-19, as most of the training and information was delivered in the cities and towns before the rural areas. They expressed their feelings:


*“As for those of us in the district and other health facilities outside the city and town, we did not have any training on COVID-19. The training was most often done in the cities and towns, neglecting those of us working in these facilities.”*
 (PA 02).


*“I did not have any training on COVID-19. All we heard was that they were training nurses and other healthcare workers at the regional level before they could come to us. However, they never came to train us. We had to rely on information on the internet.”*
 (PA 010).


**Theme 2: Stressed and burned out nurses due to increased workload.**


Participants showed that since the outbreak of the COVID-19 pandemic, nurses have suffered tremendously throughout the pandemic. They have suffered from extreme workloads, causing stress and burnout. They blamed the situation on inadequate staffing.


*“It has not been easy for all of us as nurses in this center (referring to the COVID-19 treatment center) since the outbreak of the pandemic. There is too much work here, and yet the nurses assigned to this center are not adequate, so we work around the clock. We are now tired and weak.”*
 (PA 011).

Some of the participants showed that because the government and Ministry of Health did not want to remunerate so many nurses as frontline workers in the fight against the COVID-19 pandemic, they deployed only a few nurses to the COVID-19 treatment centre, resulting in too much pressure on them.


*“We are suffering from stress and burned out here. Those of us here as nurses providing care to the COVID-19 patients are few. The government and Ministry of Health do not want to pay so many nurses as frontline workers, so they assigned only a few of us here. We are breaking down physically and psychologically.”*
 (PA 06).


*“Almost all participants interviewed in this study reported that the increased workload experienced at the treatment center affected their daily lives. Some of them described being unable to eat or sleep well at night because of the workload and stress they experienced at the COVID-19 treatment center.”*
(PA 07).


*“Since the number of critical cases of COVID-19 has increased in this centre, we, the nurses here, can no longer sleep well or eat well when getting home from work. When I come to work, I usually stand on my feet until I get close to work. I broke down physically last week.”*
 (PA 03).

Some participants threatened to resign as frontline nurses providing care to the COVID-19 patients in the treatment centre if the staffing strength of the nurses did not improve.


*“I reported to the Director of Nursing who assigned me to this centre about my intention to resign as a frontline nurse if the staffing strength of the nurses here does not improve. I am just tired and very stressed these days.”*
 (PA 08).


**Theme 3: Logistical challenges in the management of COVID-19.**


Frontline nurses reported logistical challenges, as some health facilities did not have adequate personal protective equipment (PPE) such as hand-washing items, face masks, sanitizers, and gowns. The participants had these points to say:


*“When the COVID-19 started in March 2020 in Ghana, we didn’t have enough face masks or gowns, and yet the management of the Ghana Health Service kept on telling the public that everything was available in the healthcare facilities to manage the COVID-19. I felt those were political talks because we were not having the PPEs to work.”*
 (PA 04).

A participant reported that even if the PPE was available, it was only limited to some units in the hospital and not all units.


*“The PPEs are mainly available at the outpatient departments and the isolation units, but are not available in all departments of the hospital. I work in the male medical ward, where most of the patients are admitted, yet we do not give priority to my ward in terms of the PPEs.”*
 (PA 10).

Participants showed PPE was not the only logistical challenge; there were also inadequate ventilators, oxygen, isolation rooms, and hospital beds.


*“I must say that my hospital didn’t have adequate ventilators, oxygen, isolation rooms, or hospital beds. My hospital had only two ventilators and one isolation room with only two beds. However, the number of patients we were detaining in the isolation room was always greater than the number of beds we had in the unit. This created a lot of panic and fear among health care staff in my health facility.”*
 (PA 15).


**Theme 4: Stigmatisation by family members and friends.**


Participants described being stigmatized by close relatives because they provided care to COVID-19-infected patients. Some of the frontline nurses reported having experienced stigmatization by their family members and friends because of their work providing nursing care to COVID-19 patients. They expressed their feelings:


*“It has not been easy for me since I became a frontline nurse at the COVID-19 treatment Centre. My family members and my close friends do not want to come close to me nor visit me just because of my work as a frontline nurse.”*
 (PA 15).


*“COVID-19 has caused so much stigmatization for us health workers, particularly those of us working at the COVID-19 treatment centre and the isolation units.” Since I started working at a COVID-19 treatment centre, my fiancée has refused to visit me at home for more than one month. My mother lives in town, but she told me not to come to her place to infect them since I am working with COVID-19 patients. It’s actually a worrying issue for us.”*
(PA 12).

Some of the frontline nurses showed they got frustrated and wanted to resign as frontline nurses because of the stigmatization.


*“I really wanted to resign as a frontline nurse at the initial stage of COVID-19 because of the stigmatisation from my family and friends that got me frustrated. However, I could overcome the stigma.”*
(PA 14).


*“I actually reported to my hospital nurse manager that I wanted her to replace me as a frontline nurse at the COVID-19 treatment centre because my relationship with my fiancée was on the verge of breaking down. However, later, my fiancée understood that working at the treatment centre did not mean that I was infected. Therefore, he advised me not to resign but to be very cautious of the preventive and protective measures in order not to be infected with the virus.”*
 (PA 13).

Although participants reported having been stigmatized by friends and family members, they showed there was no adequate psychological support for the nurses who provided care for COVID-19 patients at the treatment centre. Frontline nurses had to rely on their own coping mechanisms to cope with the psychological situation and stress they experienced working as frontline nurses.


*“I have to say that there was no emotional support for frontline nurses at my hospital. When I was frustrated because of the stigmatisation and the fear of being infected, I had no one to counsel or talk to. I had to find my own way of coping with my situation, which was terrible as a health worker.”*
 (PA 03).


*“There is one thing I noticed, and I feel it is not too good for us as health workers”. We offer psychological support as counselling to clients and patients when they are in any form of psychological distress. However, just imagine what we (frontline nurses) went through psychologically because of the stigmatisation and yet we were not given the psychological support.”*
(PA 01).


**Theme 5: Participants were not happy about the exclusion of some nurses from the list of frontline workers.**


The participants said they were disappointed that some nurses were excluded from the list of frontline workers and were thus not entitled to any form of reward from the government. The majority of the frontline nurses interviewed reported that until the allowances promised by the government were paid to the front-liners, no nurse knew who actually were the frontline nurses.


*“As it stands now, I do not know, and I do not think anyone knows who is a frontline nurse. This is because I do not use the criteria the Ghana Health Service used to select and describe frontline workers.”*
(PA 01).


*“The Ghana Health Service’s description of frontline workers is confusion. As I talk to you now, I do not know whether I am a frontline nurse because there are no names coming from the management of this hospital showing those who are frontline nurses and those who are not.”*
(PA 06).

Although the description of frontline nurses provided by the government through the Ghana Health Service was not clear and quite confusing, participants showed they were not pleased with the Ghana Health Service for excluding some nurses as frontline nurses. Some participants expressed their disappointment, as captured in the following quotes:


*“I must show that I am not happy that the government, through the Ghana Health Service, excluded some of our colleague nurses as frontline workers.”*
 (PA 01).


*“It amazed us that they did not include some nurses as frontline nurses, as we were told by the Ghana Health Service that not every nurse would be a frontline worker.”*
(PA 04).


*“Frontline nurses are all nurses in the health sector.” “Segregating frontline workers will amount to chaos in this difficult time.”*
(PA 02).

Participants brought up the fact that all nurses in Ghana are frontline workers because of their jobs, and that they should be treated as frontline workers and given motivation because of this. They reported their views in these quotes:


*“It surprised me that the government, through the Ghana Health Service, excluded some nurses, knowing very well that all nurses, because of their work, are frontline workers. They are the first call for every patient or client in every health facility. I am therefore surprised that they have not included every nurse as a “frontline worker”.*
 (PA 02).


*“We are not satisfied with the exclusion of some nurses as frontline workers in this COVID-19 response because all nurses are frontline workers in their units or departments of work and are there exposed to COVID-19 like any other COVID-19 frontline health worker.”*
(PA 08).

Some participants said they were told earlier by their professional union, the Ghana Registered Nurses and Midwives Association, that they would consider every nurse a frontline worker, but the government of Ghana came out to refute this claim. One participant showed:


*“As of now, I do not know who is telling the truth. Somewhere in the early part of this COVID-19 outbreak in Ghana around March 2020, we were told by the Ghana Registered Nurses and Midwives Association Executives that every nurse or midwife in active service in Ghana is considered a frontline worker, but the government through the Minister for Health said it was not true.”*
(PA 010).


**Theme 6: Recommendations for the Effective Management of COVID-19.**


Participants made various suggestions toward the prevention and management of COVID-19 in healthcare facilities. Participants suggested that the government of Ghana, through the Ministry of Health and the Ghana Health Service, needed to have made all nurses frontline workers because of their work.


*“I think what the government of Ghana, through the management of the Ministry of Health and the Ghana Health Service, needs to do is classify all nurses as frontline workers as motivation to make them put up their best work. Because nurses, because of what they do, are frontline personnel irrespective of where ever or which unit they work in the health facility.”*
(PA 14).


*“I will recommend that the government stop playing politics with nurses, who are qualified to be called frontline workers in the health facility. We classified all nurses as frontline workers. The government should rethink considering only nurses working at the COVID-19 centre and isolation units as frontline nurses. In my hospital, the nurses who were infected with COVID-19 were working in other units rather than the COVID-19 Center and isolation units.”*
(PA 09).

The participants also suggested they should provide adequate PPE to all health facilities and all units in the health facilities since all departments and units are at risk as far as COVID-19 is concerned. We captured their views in the following quotes:


*“I think one thing the government and healthcare managers need to do is ensure that they provide adequate PPEs to all health facilities in the country instead of the selective distribution of PPEs to some health facilities and some departments. It won’t help us as a country in our efforts to combat COVID-19.”*
(PA 05).


*“I think the government should do the right thing and stop deceiving the public that they have provided adequate PPEs to health workers to combat COVID-19.” “We are in dire need of PPE to work with as frontline workers.”*
(PA 07).

Some participants suggested adequate psychological support as counselling for frontline workers who are stigmatised or infected with COVID-19 to help them cope with any form of psychological distress or trauma.


*“I must say that counselling is important for us as frontline workers in combating COVID-19. All health facilities should have counsellors who will provide counselling services daily for frontline workers, especially those of us who are stigmatised or infected while providing care to COVID-19 patients and clients.”*
(PA 08).


*“I suggest counselling services should be provided and made compulsory for all frontline workers, particularly nurses. Some of the frontline nurses are suffering silently, informing no one. However, when counselling services are provided and made compulsory, it will help them cope with the stigma and psychological trauma.”*
(PA 09).

## 4. Discussion

We conducted the study to explore the experiences of frontline nurses who provided care for COVID-19 patients during the pandemic. Our findings show that most of the frontline nurses who provided nursing care for patients diagnosed with COVID-19 had limited information about COVID-19 at the onset of the pandemic because they were not given adequate training before they were deployed to the COVID-19 treatment centre to provide care. This finding is in tandem with a previous study that found that frontline nurses had limited information and insufficient knowledge of COVID-19 [27,28], and much of the information nurses received before they were deployed to provide care was a rumour that was not verified [29,30]. The novel nature of the COVID-19 pandemic meant that most healthcare providers were not familiar with the condition, let alone the management of the patients diagnosed with it. The findings of our study showed that frontline nurses who were deployed to the treatment centre needed updated and evidence-based information about how COVID-19 is transmitted and treated and how to care for patients with the disease. Providing training and resources on COVID-19 management and prevention will be critical to increasing healthcare capacity and improving the quality of nursing care.

The findings of our study illustrated that frontline nurses experienced stress and burnout because of the inadequate number of nurses deployed to the treatment centre. Similarly, previous studies have shown that many frontline nurses who provided care to COVID-19 patients experienced high levels of stress and burnout because of inadequate levels of healthcare personnel [21,22]. This finding is in line with a previous study, which reported that frontline nurses during the COVID-19 pandemic worked under unusual working conditions with increased workload, exposing them to the risk of burnout, anxiety, and stress [31]. Therefore, it is important to address the high stress and burnout among nurses, particularly during this pandemic, since health systems in developing countries such as Ghana are already overburdened with workforce shortages because of nurses’ emigration and the inequitable geographical distribution of the health workforce [32].

Furthermore, our finding shows that most of the frontline nurses were stigmatised by family members and friends because of the role they played as caregivers at the COVID-19 treatment centre. Nurses reported that although they were stigmatized by friends and family, their hospital managers did not provide them with psychological support to enable them to cope with the hazards associated with their role as frontline workers. Consistent with our findings, previous studies have reported that frontline nurses and other healthcare workers who provided care for COVID-19 patients suffered some level of stigmatisation by their families and friends because they were constantly in close contact with COVID-19 patients. Some previous studies showed that some frontline nurses during the COVID-19 pandemic were very concerned about infecting their family members and friends because of their close contact with people with COVID-19 [33,34,35,36]. These findings showed that there is a need to provide psychosocial support as professional counselling to frontline nurses deployed to provide care to COVID-19 patients in order to help them cope with stigma and avoid thoughts of resigning from the profession.

Logistical challenges emerged as one of the main findings of this study. The frontline nurses noted the dearth of resources such as the need for more personal protective equipment, ventilators, oxygen cylinders, face masks, sanitizers, gowns, isolation rooms, and hospital beds, pointing to the same dearth of resources that has plagued the whole Ghanaian healthcare system. Similar to the findings of this study, previous studies have shown that many developing countries such as Ghana faced logistical challenges such as inadequate PPE, ventilators, oxygen cylinders, face masks, gowns, and hospital beds at the initial stages of the pandemic [20,23,28,37]. Healthcare managers need to prioritise and buy basic logistics if they really want to fight and bring COVID-19 under control in a resource-constrained country such as Ghana.

Furthermore, our findings showed frontline nurses were not happy about the fact that the government of the country excluded some nurses from the COVID-19 motivation packages given to all frontline workers. Through the Ghana Health Service, the government of Ghana set up a plan for frontline workers to receive insurance and tax breaks. As an incentive to care for COVID-19 patients, frontline workers were also supposed to receive a bonus equal to 50% of their basic salary every month. However, the participants observed that most of their nursing colleagues who were not working at the treatment centre got infected with COVID-19, rather than those who worked directly with COVID-19 patients. They argued that all nurses, because of their work, are frontline workers who risk their lives daily to help fight the COVID-19 pandemic and therefore deserve to be motivated and not just a few nurses, as opposed to the action taken by the Ghana Health Service. Consistent with our findings, previous studies showed that nurses have deep concerns about the inability of the government and other stakeholders to provide motivational incentives such as allowances, tax waivers, and insurance packages for all nurses since every nurse is at risk of being infected with COVID-19, irrespective of the unit or area of practice [38,39]. The above finding implies that in future epidemics, the government should improve nurses’ working conditions to improve patient care quality.

Finally, the participants made some suggestions for stakeholders in the fight against COVID-19. The recommendations were mainly to address the challenges that confronted frontline nurses. For the effective control and management of the COVID-19 pandemic, there was a need for the Ministry of Health and Ghana Health Service to have included all nurses as frontline workers because of their work. They also suggested that there was a need for the government to provide adequate PPE in all health facilities and adequate psychological support as counselling for frontline workers who are stigmatised or infected with COVID-19 to help them cope with any form of psychological distress or trauma. Consistently with the literature, several previous studies recommended that to effectively fight against COVID-19, there was a need to provide adequate motivation for frontline healthcare workers and adequate provision of basic logistics such as PPE, ventilators, oxygen cylinders, and hospital beds [39]. Providing adequate psychological and social support for frontline healthcare workers providing nursing care during the COVID-19 pandemic was recommended by previous studies [40]. Frontline nurses who provided nursing care during the COVID-19 pandemic experienced psychological problems caring for COVID-19 patients, and therefore, there was a need for healthcare organisations to provide them psychological support to mitigate the anxiety and stress frontline workers experience caring for COVID-19 patients.

## 5. Limitations of the Study

The study has some limitations. We conducted this study among frontline nurses who provided care to people diagnosed with COVID-19 in the primary treatment centre for COVID-19 patients, in the Volta Region of Ghana. This limits the ability of this study to generalize its findings. Another limitation of this study was that the population in our study was limited to the experiences of only frontline nurses who provided care to people who were diagnosed with COVID-19. We could not explore the experiences of other frontline healthcare workers, such as doctors, who also provided direct care for COVID-19 patients and were not included in this study. We recommend further studies be conducted to explore the experiences and challenges of other frontline healthcare workers who provided care during the COVID-19 pandemic across all treatment centres in Ghana to have a broader and more comprehensive understanding of the experiences and challenges of frontline healthcare workers during the COVID-19 pandemic in Ghana. These limitations notwithstanding, this study provided an in-depth understanding of the experiences and challenges of frontline nurses who provided care to COVID-19 patients during the pandemic in Ghana.

## 6. Conclusions

This study gave us a deep understanding of what frontline nurses who cared for COVID-19 patients during the pandemic went through. Our study found that the frontline nurses experienced both physical and psychological problems caring for COVID-19 patients at the treatment centre. Many frontline nurses experienced several challenges, such as inadequate information on COVID-19 prevention and management in the early stages of the pandemic, logistical inadequacy, and stigmatisation, providing care for COVID-19 patients during the pandemic affecting quality nursing care, work productivity, and efficiency. Therefore, there is a need for nurse managers to provide support for frontline nurses providing care for patients with COVID-19. We recommend that frontline nurses be provided with physical and psychosocial support as counselling by psychologists to cope with the stress and stigma associated with the COVID-19 pandemic.

## 7. Implications for Nursing Management

The findings of this study show that nurse managers and leaders are required to play an important role in providing support to their nurses during and after the COVID-19 pandemic. Nurse managers are supposed to ensure that nurses providing frontline care to COVID-19 patients are well trained and equipped with the requisite information to enable them to provide care effectively to COVID-19 patients without the risk of infecting themselves or their families. The findings of this study also imply that nurse managers are supposed to play an advocacy role by ensuring that government and healthcare managers provide nurses who provided care during the pandemic with good working conditions, adequate resources, and motivation to perform their role effectively and efficiently.

The results of this study also show that the nursing administration needs to make sure that experts provide psychosocial support to frontline nurses during epidemics. This will help them deal with the stress and stigma that comes with being a frontline nurse during the COVID-19 response. Healthcare managers and nursing managers providing financial, psychological, and social support to nurses can go a long way towards preventing nurses experiencing thoughts of quitting their profession in times of epidemics such as the COVID-19 pandemic. It is therefore imperative that future organizational and government strategies and interventions improve the conditions of services and the well-being of the frontline nurses providing care during the pandemic.

## Figures and Tables

**Table 1 healthcare-11-01028-t001:** Profile of participants.

Participant Code	Rank in Nursing	Age	Sex	Marital Status	Work Experience
P01	Senior Nursing Officer	32	Female	Married	10
PA02	Senior Nursing Officer	40	Female	Married	16
PA03	Nursing Officer	28	Female	Married	5
PA04	Senior Nursing Officer	29	Male	Married	7
PA05	Senior Nursing Officer	30	Female	Married	9
PA06	Senior Nursing Officer	45	Female	Married	17
PA07	Nursing Officer	27	Male	Single	4
PA08	Principal Nursing Officer	50	Female	Married	25
PA09	Nursing Officer	35	Female	Married	13
PA10	Nursing Officer	34	Female	Married	9
PA11	Nursing Officer	26	Male	Single	5
PA12	Senior Nursing Officer	42	Female	Married	20
PA13	Principal Nursing Officer	47	Female	Married	17
PA14	Nursing Officer	27	Male	Single	5
PA15	Nursing officer	26	Male	Single	4

## Data Availability

The datasets used for the analyses of the current study are available from the corresponding author on request.

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
