# Peer review of "A Qualitative Study on Frontline Nurses’ Experiences and Challenges in Providing Care for COVID-19 Patients in the Volta Region of Ghana: Implications for Nursing Management and Nursing Workforce Retention"

_healthcare, 2023, doi:10.3390/healthcare11071028_

Round 1

Reviewer 1 Report

There some minor issues:

Since there are several qualitative studies investigating the Nurses' experiences, please indicate in the literature, why Ghanaian context is important for studying the same topic 

I think this study is a kind of case study rather than a descriptive qualitative research.

Please indicate which coding techniques or methods were applied?

Please explain how many coders coded the themes. If there were more than one please explain how you maintained the Intercoder Reliability. Othervise expleain realibility and validity issues.

In Table 1, please add experince of the participants. Since there is no discussion about based on the ontained results and marital status of the participants, I could not understand why this information was included

Please add a table for the main themes and sub themes with their frequencies

Author Response

RESPONSE TO REVIEWER 1 COMMENTS

Point 1: Since there are several qualitative studies investigating the Nurses' experiences, please indicate in the literature, why Ghanaian context is important for studying the same topic 

Response 1: The few studies that explored the experiences of frontline healthcare workers who provided care during the COVID-19 pandemic were conducted outside the Ghanaian context]. Therefore, there is a need to conduct a study on the experiences and challenges of frontline nurses in Ghana who are providing care for COVID-19 patients. This study will provide a platform for nurses to share their experiences, challenges, and insights on providing care for COVID-19 patients. It will also allow for the identification of the specific challenges that nurses face and how they can be addressed to improve the quality of care provided for COVID-19 patients in Ghana.

Point 2: I think this study is a kind of case study rather than descriptive qualitative research.

Response 2: This study adopted a case study research design to collect data. Case study design is a research method that involves the detailed examination of a particular case, such as an individual, group, or organization, in order to gain a deep understanding of a phenomenon

Point 3: Please indicate which coding techniques or methods were applied?

Response 3: The researchers adopted an open-coding approach, where they read and examined the data to identify concepts and themes.

Point 4: Please explain how many coders coded the themes. If there were more than one, please explain how you maintained the Intercoder Reliability.

Response 4: Three authors, who have PhDs in nursing and experience in qualitative research methods, manually conducted the data analysis. They did this to ensure the trustworthiness of the analysis, as using multiple coders ensures intercoder reliability.

Point 5: In Table 1, please add experience of the participants

Response 5: The experience of the participants has been added to Table 1

Reviewer 2 Report

Even though many similar studies have been conducted regarding COVID-19, this study "A qualitative Study on Frontline Nurses’ Experiences and Challenges of Providing Care For COVID‐19 Patients in the Volta Region of Ghana. Implications for Nursing Management and Nursing Workforce Retention" was new of its  kind in Ghana.

Implications for Nursing Management provide insight into the topic.

I have some suggestions:

- In data analysis section, "Three authors, who have PhDs in nursing with experience in qualitative research methods, did manually conducted the data analysis". Change it to "Three authors, who have PhDs in nursing with experience in qualitative research methods, manually conducted the data analysis".

- You mentioned that majority10 (67%) of the participants were female nurses, while 5 (33%) were nurses. Write as 5 were male nurses. The general term nurse will be mostly used for female nurses.

- In your study, majority (11) of the participants were married while only 4 (27%) were never married. What is the association of this marital status with your study?

- The 6 themes could be presented in the form of table for easy understanding and to avoid boring for the readers to read the vast text. 

Author Response

Response to Reviewer 2 Comments

Point 1: In data analysis section, "Three authors, who have PhDs in nursing with experience in qualitative research methods, did manually conducted the data analysis". Change it to "Three authors, who have PhDs in nursing with experience in qualitative research methods, manually conducted the data analysis".

Response 1: This has been changed. Three authors, who have PhDs in nursing and experience in qualitative research methods, manually conducted the data analysis. They did this to ensure the trustworthiness of the analysis, as using multiple coders ensures intercoder reliability

Point 2: You mentioned that majority10 (67%) of the participants were female nurses, while 5 (33%) were nurses. Write as 5 were male nurses. The general term nurse will be mostly used for female nurses

Response 2: This has been addressed. The results showed that the majority of 10 (67%) of the participants were female nurses, while 5 (33%) were male nurses.

Point 3: In your study, majority (11) of the participants were married while only 4 (27%) were never married. What is the association of this marital status with your study?

Response 3: This is a qualitative study and so, authors did not examine the association of this marital status with your study. The marital status of participants is to inform the reader, the demographic profile of the participants